# Effects of Psychological Interventions on Performance Anxiety in Performing Artists and Athletes: A Systematic Review with Meta-Analysis

**DOI:** 10.3390/bs13110910

**Published:** 2023-11-07

**Authors:** Marc Niering, Teresa Monsberger, Johanna Seifert, Thomas Muehlbauer

**Affiliations:** 1Institute of Biomechanics and Neurosciences, Nordic Science, 30173 Hannover, Germany; niering@nordicscience.de (M.N.); seifert.johanna@mh-hannover.de (J.S.); 2Institute of Neuroscience and Psychology, University of Glasgow, Glasgow G12 8QQ, UK; 3Department of Psychiatry, Social Psychiatry and Psychotherapy, Medical School Hannover, 30625 Hannover, Germany; 4Division of Movement and Training Sciences/Biomechanics of Sport, University of Duisburg-Essen, 45141 Essen, Germany

**Keywords:** professional sports, performing arts, state performance anxiety, trait performance anxiety, psychological interventions

## Abstract

Levels of state and trait anxiety are relevant for performing artists and professional athletes to obtain optimal performance outcomes. However, evidence-based knowledge regarding the effectiveness of psychological interventions on performance anxiety is currently minimal. Thus, the objective of this systematic review with meta-analysis was to characterize, aggregate, and quantify intervention effects on measures of state and trait performance anxiety in performing artists and professional athletes. A systematic search of the literature according to the PRISMA guidelines was conducted on the databases PubMed, Medline, SPORTDiscus, PsycInfo, Embase, and Web of Science from 1 January 1960 to 9 November 2022. The search only included controlled studies employing pre–post measures and excluded performing arts fields that do not depend on fine motor skills. Initially, 1022 articles were identified; after removing duplicates and assessing abstracts and full texts, 20 articles were used to calculate weighted standardized mean differences (*SMDs*). In terms of state performance anxiety, our results revealed a large overall effect (*SMD* = 0.88), a medium effect (*SMD* = 0.62) for studies using scales with total scores (i.e., MPAI-A, STAI), and large effects (cognitive anxiety: *SMD* = 0.93, somatic anxiety: *SMD* = 0.92, self-confidence: *SMD* = 0.97) for studies applying scales with sub-scores (i.e., CSAI-2R)—all in favour of the intervention groups. Regarding trait performance anxiety (e.g., SCAT), we detected a small effect (*SMD* = 0.32), also favouring the intervention groups. Interventions to reduce performance anxiety in performing artists and professional athletes revealed varying levels of effectiveness ranging from small (trait) to large (state). Therefore, future studies should investigate modalities to increase intervention efficacy, especially for the small-sized changes in trait performance anxiety.

## 1. Introduction

Anxiety-related mental health problems such as social anxiety disorder or specific phobias are amongst the most prevalent mental health disorders in the world [1]. Among individuals participating in professional sports or performing arts, the prevalence of anxiety-related disorders varies between 19.5% [2] and 34% [3]. The aetiology of these anxiety-related mental health problems, especially performance anxiety, is comparatively vague. While the cause of the phenomenon fear is clearly linked to identifiable objects or circumstances (e.g., spiders or darkness), anxiety is often not so clearly linked to a distinct cause. A person experiencing performance anxiety tends to act in anticipation of a possible danger, therefore starting a pre-encounter defence (i.e., a response toward the possibly dangerous circumstances where a typical “threat” has yet to be encountered) which is associated with activation of the prefrontal cortex [4].

Performance anxiety typically manifests psychosomatically as somatic and cognitive symptoms which vary depending on the individual [5], the situation, and athletic or artistic discipline [6]. The somatic response includes an increased heart rate, trembling, breathing difficulty, and excessive sweating. Cognitive symptoms are characterized by an increased outcome-related worry, overthinking, and self-oriented cognitions, often resulting in attentional disruption [7]. Furthermore, altered behaviour is evident in the form of overt and covert avoidance, affecting both quality of life and athletic performance [8]. Performance-related anxiety can be classified as either state or trait anxiety. State anxiety is a response to a perceived threatening situation, while trait anxiety is an inherent tendency towards anxiety that may not necessarily be triggered by external stimuli [9].

When individuals are under pressure to perform at their best, they are at times unable to deliver their best performance. This is often due to performance anxiety, which arises when the pressure evoked by performative or competitive stressors is perceived as threatening [10]. Most frequently, performance anxiety occurs when the affected individual perceives an imbalance between the demands placed on them and their ability to fulfil these [11]. This also leads to an emotion-related decline in mood state, which, combined with psychological stress, leads to impaired performance [12]. According to Willis et al. [13], performing artists include a wide range of disciplines that focus on the mastery of techniques and artistic expression in front of an audience. These include musicians, dancers, actors, comedians, and circus artists. The perhaps most debilitating aspect of performance anxiety for performing artists is cognitive interference causing the performers to lose control over previously mastered movements [14]. Compared to other artists, performing artists are more predisposed to suffering from performance anxiety due to an interplay of various occupational factors [15,16]. In ballet, a combination of fine motor and gross motor skills constitutes the complex movement combinations executed by the performers [17], which serve as a vehicle or language for expressing a personal emotional state to the viewer [18]. Currently, the available literature suggests that performing artists experience performance anxiety not only when performing on stage, but also in rehearsal or regular classes. Barrell and Terry [19] found similar levels of trait anxiety in dance students and professional dancers, with neither performance level nor sex showing significant differences. Performance anxiety was shown to impair performing artists’ motor performance and, if experienced over an extended period paired with insufficient coping skills, overall mental wellbeing is also impacted, leading to depression and clinical anxiety in severe cases [15]. Concurrent with previous research in sports and the performing arts, the relevant literature [6,14,20] concluded that cognitive anxiety appears to have a greater effect not only on performing artists’ perceptions of the direction of anxiety symptoms but also on their actual physical performance.

According to Tomporowski and Pesce [21], athletes and performing artists face increasing cognitive, psychological, and physical demands due to the rising skill levels in their respective fields. These demands entail recognizing and adapting to sudden situations, as well as learning specific techniques while upholding a high level of creative effort. To fulfil these requirements, athletic and artistic demands such as athleticism, flexibility, virtuosity, and, moreover, individuality, creativity, and artistry are imposed on performing artists. Similar to athletes in sports, this broad array of demands requires musicians, actors, and dancers to devote themselves to intensive training to meet and exceed these standards, generally from early childhood onwards [22,23,24]. Inevitably, the commitment to their art form from a young age and throughout adolescence frequently paves the way for a fusion of artists’ occupational identity with their self-identity [25]. The perceived threat to a personal identity associated with occupation-related failure lies at the root of performance anxiety, thus contributing to an increased likelihood of its occurrence and greater intensity of symptoms [19,26,27]. Furthermore, a sports identity characterized by negative affectivity, being a world-class performer, or being female also increases the level of performance anxiety [28].

However, athletic disciplines relying predominantly on gross motor skills, such as running, might benefit from the energy rush experienced by increased sympathetic arousal facilitating performance [29]. In contrast, this sympathetic overactivation might be perceived as debilitating by golfers and musicians as well as any athletes or performing artists relying on fine motor skills due to its potential to impede motor skill accuracy. Specifically, it could impair motor accuracy during these movements, as fine motor skills require many more cognitive resources [10,30]. This conscious effort frequently causes expert performers to perform significantly below their usual level, resulting in what scholars have coined “choking under pressure” [30,31]. Therefore, fine motor control, combined with motor sequencing, can be considered a common risk factor for performance anxiety in sports and the performing arts, since it plays a crucial role in determining performance, thus making these groups comparable [32]. Furthermore, this population not only illustrates similarities in the underlying neurophysiological mechanisms causing impaired motor performance [33], but also comparable physiological responses which cause the somatic symptoms of anxiety [34].

Research has investigated diverse approaches to support athletes and performing artists in overcoming performance anxiety’s debilitating effects. Many of these methods are generally assigned to the domain of psychological interventions. They range from relaxation techniques, slow breathing, bio- and neurofeedback, self-talk, cognitive restructuring, mindfulness, and acceptance and commitment therapy (ACT) to virtual reality exposure, hypnotherapy, and somatic techniques [35,36,37,38,39,40]. These mental skill techniques have been developed, tested, and successfully implemented into practice to improve coping with physiological overarousal and cognitive interference in pursuit of athletic and musical performance optimization. To improve an individual’s ability to cope with anxiety symptoms, interventions typically focus on reducing sympathetic activation, changing the appraisal or direction of anxiety symptoms, restructuring, and improving negative thoughts that can impact performance [41].

Nevertheless, research investigating effective methods to alleviate performance anxiety by taking the specific demands of performing artists into account is sparse [13,42,43], despite the apparent prevalence of performance anxiety within this field. Performing artists commonly employ maladaptive coping strategies, and coping skills are rarely formally taught [19,25,44] or assessed for adherence [45], making intervention practicality determinations rare. Thus, it becomes clear that there is a need for evidence-based results in the field of psychological interventions. This current research gap serves as the rationale for this paper. This systematic review and meta-analysis assesses the current state of the art on the effects of psychological interventions used in sports and the performing arts to reduce performance anxiety and to determine their effects on performance anxiety in performing artists and athletes. It aims to provide relevant insights for the treatment and prevention of performance anxiety in the performing arts. In addition, promising research questions from related scientific evidence will be identified to encourage and facilitate interdisciplinary approaches.

## 2. Materials and Methods

In this meta-analysis, we followed the recommendations of the Preferred Reporting Items for Systematic Reviews and Meta-Analysis (PRISMA) statement guidelines search (see Appendix A) [46].

### 2.1. Literature Search

A systematic computerized search for relevant empirical studies was performed in PubMed, Medline, SPORTDiscus, PsycInfo, Embase, and Web of Science using the following Boolean search strategy: “(performance OR sport OR psychological) AND (anxiety) AND (intervention OR training) AND (music OR artists OR athletes OR sports)”. Further, the search was limited to the following: full-text availability, publication dates: 1 January 1960 to 9 November 2022, language: English, article type: no review. Moreover, the reference lists of the included articles were screened to identify other suitable studies for inclusion in our analysis.

### 2.2. Selection Criteria

To be eligible for inclusion in our systematic review with meta-analysis, we considered studies if they provided enough relevant information regarding the PICOS (Population, Interventions, Comparators, Outcomes, Study design) approach. We used the following criteria to determine eligibility: (a) Population: athletes or performing artists depending on fine motor skill performance including a technical component; (b) Intervention: psychological interventions; (c) Comparator: active or passive control group (i.e., different psychological intervention, no training at all); (d) Outcome: at least one measure of anxiety; (e) Study design: controlled trials with pre- and post-measures. The exclusion criteria were as follows: (a) participants’ discipline was not dependent on fine motor skill performance or without a technical component (e.g., actors); (b) reported data did not allow for calculation (i.e., no central tendency and dispersion measure provided in the Section 3 or upon request); (c) effects were examined without control condition; (d) assessments did not include a psychological outcome measure; (e) cross-sectional study design. Three independent reviewers (M.N., T.Mo., T.Mu.) assessed the eligibility of the relevant papers by analysing the titles, abstracts, and full texts of the respective articles.

### 2.3. Methodological Study Quality

Each article was evaluated according to the Scottish Intercollegiate Guidelines Network Methodology checklist for randomized controlled trials [47] by two independent authors (M.N., J.S.) in order to assess the quality of the eligible articles and reduce risk of bias (see Appendix A). The possible classifications are low quality (−), acceptable quality (+), and high quality (++). Studies classified as unacceptable (0) were rejected. The degree of agreement between the two assessors was 91%. When disagreement between assessors occurred, a consensus meeting was performed, and an additional rating was obtained from a third assessor (T.Mu.) to achieve consensus.

### 2.4. Synthesis of Results

In the category of state performance anxiety, the Competitive State Anxiety Inventory 2 Revised score (CSAI-2R) was identified as the most common test item and illustrated high factorial [48] and subscale [49] validity. In terms of trait performance anxiety, the Sport Competition Anxiety Test (SCAT) is the most used instrument and reports high internal consistency [50]. If studies used other outcomes that could be assigned to one of the two categories, these were presented as alternative outcomes in Table 1.

Further, we considered the use of different psychological coping skills during an intervention. Treatment modality was coded according to the following parameters: training weeks/sessions and exercise duration. If the considered studies did not disclose relevant results, the authors were contacted via email [51,52]. When authors failed to respond to our request, or the requested data were no longer available [51,52], we excluded the respective outcome measure.

### 2.5. Statistical Analysis

To quantify the effects of psychological interventions on performance anxiety in athletes or performing artists, the within-subject standardized mean difference was calculated as *SMD_W_* = (pretest mean value—post-test mean value)/pretest standard deviation and the between-subject standardized mean difference as *SMD_b_* = (post-test mean value in the experimental group—post-test mean value in the control group)/pooled standard deviation [53] using Review Manager version 5.4.1 (i.e., random-effects model). In addition, included studies were weighted according to the magnitude of the respective standard error. *SMD_W_* and *SMD_b_* can be positive or negative. Positive *SMD_W_* values indicate an improvement in performance anxiety (e.g., increase in STAI) from pretest to post-test, while negative *SMD_W_* values indicate a decrease in performance anxiety (e.g., decrease in STAI). Positive *SMD_b_* values indicate an improvement in performance in favour of the intervention group (INT), while negative values indicate an improvement in favour of the control group (CON). *SMD_W_* and *SMD_b_* values can be classified and interpreted according to Cohen [54] into the following ranges: 0 ≤ 0.49 representing small effects, 0.50 ≤ 0.79 representing moderate effects, and ≥0.80 representing large effects. Further, heterogeneity (*I*^2^) was computed by using the formula provided by Deeks et al. [55]: *I*^2^ = (*Q* − *df/Q*) × 100%, where *Q* is the chi-squared statistics and *df* represents the degrees of freedom [56]. According to Deeks et al. [55], heterogeneity can be interpreted as trivial (0 ≤ 40%), moderate (30 ≤ 60%), substantial (50 ≤ 90%), or considerable (75 ≤ 100%). In addition, a separate (state/trait) qualitative funnel plot evaluation was performed to examine a potential publication bias.

## 3. Results

### 3.1. Selection of Studies

Figure 1 illustrates the search strategy and selection process for the psychological interventions. In the databases PubMed, Medline, SPORTDiscus, PsycInfo, Embase, and Web of Science, a total of 1006 articles for psychological interventions were identified for further consideration. A manual search of reference lists identified another 16 articles. After removing duplicates, excluding articles based on title or abstract, as well as reviews, case studies and experimental study designs, 98 articles remained. In the full-text search, 78 articles were excluded based on the selection criteria. Thirty-six studies examined a study population not relying on fine motor skills (e.g., actors), or a mixed population including those. Another 33 studies did not use a psychological intervention, seven did not report any psychological outcome measures, and two did not provide conclusive data.

### 3.2. Study Characteristics

The characteristics of the included studies are listed in Table 2 and illustrate the authors, the year of publication, the participant characteristics, the intervention and control groups, the details of the interventions conducted, the test procedures, and outcome measures, as well as the results of the individual groups.

### 3.3. Participant Characteristics

A total of 707 participants were examined in the studies included in this analysis. The participants were all without diagnosed mental diseases and between 10 and 57 years old. Five studies investigated a mean age of 10.3–15.8 years [20,57,58,59,60], two studies of 14.95–18.35 years [61,62], two of 17.64–23.0 years [50,63,64,65], eight of 18.1–29.9 years [49,66,67,68,69,70,71], and one record studied 26.2–57.8 year olds [72]. Two studies examined only female participants [57,66], six only male participants [61,62,64,65,67,73], one study examined 29 women and 4 men [72], and all others had a more balanced male-to-female ratio. For participants’ sports or professions, seven studies illustrated results for musicians [20,49,57,68,69,71,72], two for swimmers [50,61], two for tennis players [58,60], two for gymnasts [59,63], two for basketball players [64,66], and one each for rugby players [65], karate athletes [70], wushu athletes [73], soccer players [62], and golfers [67].

**Table 2 behavsci-13-00910-t002:** Effects of psychological interventions on state and trait performance anxiety in performing artists and athletes.

References	No. of Subjects (Sex); Age (Mean ± SD, or Range); Activity	Groups; Type of Intervention	Intervention: No. of Training Weeks/Sessions; Single Session Duration	Test Modality; Outcome Measures	Results, Mean (SD)	Level of Evidence
Braden et al. [57]	62 (62 F); 14 ± 0.9 y; music students	INT (*n* = 30): Multimodal CON (*n* = 32): No intervention	INT: 8 w/8 sessions; group-based; psychological skills training. S1: Peak performance, personal strengths. S2: Goal setting, motivation. S3: Self-talk and affirmations. S4: Routines, relaxation, self-talk. S5: Mental practice techniques. S6: Stress management. S7: Flow. S8: Resilience, coping, positive thinking. CON: No intervention	MPAI-A state (score). State performance anxiety. Pre-/post-intervention.	MPAI-A state: INT pre: 28.69 (10.12) CON pre: 26.69 (10.60) INT post: 18.90 (8.44) CON post: 25.63 (9.80)	++
Clark & Williamon [49]	23 (14 F, 9 M); 24 ± 5.9 y; undergraduate music students	INT (*n* = 5): Multimodal CON (*n* = 9); No intervention	INT: 9 w/18 sessions, group-based; 60 min + individual; 30 min; theoretical introduction of mental skills, subsequent practice. w 1–3: Goal setting, effective practice, time management. w 4–6: Relaxation and arousal control, relaxation techniques, arousal regulation through cognitive restructuring and self-talk. w 7–9: Mental rehearsal, imagery, performance preparation and analysis. CON: No intervention.	CSAI-2R (score). State performance anxiety. STAI (score). Trait Performance anxiety. Pre-/post-intervention.	CSAI-2R: Cognitive Anxiety: INT pre: 11.29 (3.89) CON pre: 11.67 (4.09) INT post: 10.36 (3.77) CON post: 11.11 (3.98) Somatic anxiety: INT pre: 13.71 (3.99) CON pre: 12.44 (3.43) INT post: 11.86 (3.32) CON post: 10.22 (3.56) Self-confidence: INT pre: 12.36 (2.71) CON pre: 11.22 (2.73) INT post: 13.21 (3.79) CON post: 11.33 (2.35) STAI trait: INT pre: 45.65 (9.12) CON pre: 49.00 (8.12) INT post: 41.43 (7.65) CON post: 43.44 (7.92)	+
Georgakaki & Karakasidou [50]	44 (21 F, 23 M); 19.8 ± 2.16 y; competitive swimmers	INT (*n* = 23): Motivational self-talk CON (*n* = 21): No intervention	INT: 3 w/3 sessions; group-based; 90 min; self-talk, emotional regulation, thought-stopping, motivational cues. Implementation into sports practice. CON: No intervention.	SCAT (score). Trait Performance anxiety. Pre-/post-intervention.	SCAT: INT pre: 23.22 (±3.67) CON pre: 24.05 (±4.73) INT post: 20.39 (±4.04) CON post: 23.81 (±3.92)	+
Hatzigeorgiadis et al. [58]	72 (36 F, 36 M); 13 ± 1.8 y; competitive tennis players	INT (*n* = 54): Motivational self-talk CON (*n* = 18): Lectures on tactical aspects in tennis	INT: 5 sessions; group-based; S1: Baseline trial. skill assessment under artificially induced stressful conditions, subsequent completion of CSAI-2R. S2–4: Introduction to self-talk. self-talk practice, use of self-talk cues (instructional and motivational) when playing. participants were asked to indicate on a 10-point scale how frequently they used cues. S5: Post-intervention test. stressful situation as in pretest. INT state a motivational cue they would use. CON: 5 sessions; group-based. S1: Identical with INT. S2–4: Lectures on tactical aspects. S5: Identical with INT.	CSAI-2R (score). State performance anxiety. Pre-/post-intervention.	CSAI-2R: Cognitive anxiety: INT pre: 1.15 (0.68) CON pre: 1.25 (0.72) INT post: 0.89 (0.65) CON post: 1.38 (0.80) Somatic anxiety: INT pre: 0.83 (0.57) CON pre: 0.90 (0.59) INT post: 0.61 (0.45) CON post: 0.90 (0.76) Self-confidence: INT pre: 1.59 (0.70) CON pre: 1.59 (0.75) INT post: 1.89 (0.63) CON post: 1.59 (0.78)	++
Hoffman & Hanrahan [72]	33 (29 F, 4 M); 42 ± 15.8 y; musicians	INT (*n* = 15): Multimodal CON (*n* = 18): No intervention	INT: 3 w/3 sessions; group-based; 60 min. S1: Introduction to emotional regulation, cognitive appraisal, ideal activation state. S2: Identification of individual debilitative thought patterns, learning and practice of self-talk regulation. S3: Cue-controlled self-talk, guided imagery. CON: No intervention.	STAI (score). State performance anxiety. PAI (score). State performance anxiety. Pre-/post-intervention and 4 w post-intervention.	STAI State: INT pre: 39.13 (8.37) CON pre: 44.44 (8.41) INT post: 39.00 (8.38) CON post: 41.61 (11.44) PAI: INT pre: 54.47 (12.06) CON pre: 51.89 (13.17) INT post: 50.07 (9.00) CON post: 53.94 (11.65) INT post2: 47.25 (8.58)	+
Marshall & Gibson [59]	19 (13 F, 6 M); 13 ± 2.7 y; acrobatic gymnasts	INT (*n* = 11): Imagery CON (*n* = 8): No intervention	INT: 4 w/4 sessions; group-based; 15 min; guided sessions preceding regular physical training. Homework task assigned to participants: elaborate and personalize imagery experience. CON: No intervention.	CSAI-2 (score). State performance anxiety. Pre-/post-intervention.	CSAI-2: Cognitive anxiety: INT pre: 19.55 (4.59) CON pre: 19.63 (4.37) INT post: 17.82 (4.66) CON post: 20.88 (7.18) Somatic anxiety: INT pre: 20.64 (6.86) CON pre: 20.13 (5.46) INT post: 18.55 (6.62) CON post: 20.00 (7.95) Self-confidence: INT pre: 21.09 (4.30) CON pre: 21.13 (1.96) INT post: 26.18 (4.38) CON post: 22.13 (3.91)	+
Osborne et al. [20]	23 (14 F, 9 M); 14 ± 1.2 y; music students	INT (*n* = 14): Multimodal CON (*n* = 9): Behaviour exposure only	INT: 2 w/7 sessions, 3 group-based; 60 min + 4 individually; 45 min; goal-setting, relaxation techniques, cognitive restructuring, self-talk. The different techniques were each introduced and practiced in several sessions. Homework tasks included in intervention. Behavioural exposure through pre- and postintervention solo performances. CON: No intervention. Behavioural exposure through pre- and postintervention solo performances.	MPAI-A trait (score). Trait Performance anxiety. Pre-/post-intervention.	MPAI-A trait: INT pre: 68.1 (9.7) CON pre: 65.2 (7.5) INT post: 50.0 (13.4) CON post: 52.0 (14.2)	−
Terry et al. [60]	100 (58 F, 42 M); 14 ± 1.8 y; elite junior tennis players	INT (*n* = 25): Relaxation INT2 (*n* = 24): Mental rehearsal INT3 (*n* = 25): Relaxation + mental rehearsal CON (*n* = 26): Concentration grid exercise	INT1: 1 session; group-based; 10 min; relaxation. INT2: 1 session; group-based; 15 min; Mental rehearsal. INT3: 1 session; group-based; 10 min; relaxation + 15 min; group-based; Mental rehearsal. CON: 1 session; group-based; 1 min; concentration grid exercise.	CSAI-2 (score). State performance anxiety. Pre-/post-intervention	CSAI-2: Cognitive Anxiety: INT1 pre: 14.7 (4.0) INT2 pre: 14.9 (3.5) INT3 pre: 16.6 (4.3) CON pre: 14.0 (3.1) INT1 post: 12.0 (3.9) INT2 post: 12.6 (3.4) INT3 post: 14.5 (5.2) CON post: 13.2 (2.9) Somatic Anxiety: INT1 pre: 19.5 (4.9) INT2 pre: 19.3 (4.5) INT3 pre: 20.3 (5.4) CON pre: 17.9 (3.8) INT1 post: 17.4 (5.1) INT2 post: 16.6 (4.8) INT3 post: 18.5 (6.1) CON post: 17.4 (5.3) Self-confidence: INT1 pre: 27.2 (4.4) INT2 pre: 26.3 (5.0) INT3 pre: 25.8 (5.1) CON pre: 26.8 (5.2) INT1 post: 30.0 (3.8) INT2 post: 28.5 (6.1) INT3 post: 29.0 (5.4) CON post: 27.2 (5.6)	+
Thurber et al. [69]	14 (5 F, 9 M); M = 23 y; music students	INT (*n* = 7): HRV biofeedback CON (*n* = 7): No intervention	INT: 3 w/5 sessions; individual; 30–50 min; biofeedback-sessions. Instruction in the concepts of physiological arousal, emotional memory, nervous system, relaxed breathing. Additional, related reading material provided. Individual coaching regarding personal, MPA-related challenges. CON: No intervention.	STAI (Score). State performance anxiety. PAI (Score). State performance anxiety. Pre-/post-intervention.	STAI State: INT pre: 39.42 (11.67) CON pre: 41.28 (11.33) INT post: 31.85 (7.75) CON post: 39.41 (6.46) PAI: INT pre: 46.00 (14.69) CON pre: 38.14 (8.39) INT post: 44.42 (15.61) CON post: 38.35 (8.09)	−
Whitaker [71]	18 (10 F, 8 M); M = 26 y; university music students	INT (*n* = 9): Multimodal CON (*n* = 9): No intervention	INT: 6 w/3 sessions; group-based and individual. S1: Introduction to the intervention, explanation of human stress responses and cognitive appraisal. PMR tape distributed for daily use. S2: The power of thought. Constructive analysis of own videotaped performance. MR + autogenic training tape distributed for daily use. S3: Self-talk regulation, goal-setting. PMR + autogenic training tape distributed for daily use. CON: No intervention.	STAI (score). State performance anxiety. Pre-/post-intervention and 6 w post-intervention.	STAI State: INT pre: 50.78 (12.02) CON pre: 51.78 (9.94) INT post: 36.22 (9.48) CON post: 48.11 (9.90) INT post2: 35.33 (9.40) CON post2: 43.67 (12.17)	−
Wolch et al. [64]	32 (32 M); 21 ± 2 y; basketball players	INT (*n* = 16): Mindfulness CON (*n* = 16): Behaviour exposure only	INT: 1 session; group-based; 15 min; guided mindfulness meditation. CON: 1 session; group-based; 15 min; listening to basketball history.	CSAI-2R (score). State performance anxiety. Pre-/post-intervention.	CSAI-2R: Cognitive Anxiety: INT pre: 15.9 (5.6) CON pre: 16.6 (5.8) INT post: 15.0 (3.9) CON post: 17.9 (4.0) Somatic Anxiety: INT pre: 14.7 (4.9) CON pre: 16.8 (5.1) INT post: 14.2 (3.0) CON post: 18.6 (5.9) Self-confidence: INT pre: 30.6 (5.1) CON pre: 33.8 (4.6) INT post: 31.3 (4.6) CON post: 33.9 (5.4)	++
Yahya et al. [65]	58 (58 M); 21 ± 2 y; rugby players	INT (*n* = 29): Imagery CON (*n* = 29): No intervention	INT: 6 w/18 sessions; individual; 2 min; imagery during kicking practice. CON: 6 w/18 sessions; individual; 2 min; kicking practice without imagery.	CSAI-2R (score). State performance anxiety. Pre-/post-intervention.	CSAI-2R: Cognitive Anxiety: INT pre: 14.8 (3.0) CON pre: 14.7 (2.6) INT post: 11.9 (2.1) CON post: 14.3 (1.6) Somatic Anxiety: INT pre: 18.2 (5.1) CON pre: 18.2 (3.8) INT post: 14.0 (2.5) CON post: 17.8 (2.7) Self-confidence: INT pre: 13.3 (2.4) CON pre: 13.3 (3.5) INT post: 16.1 (1.7) CON post: 13.8 (2.6)	+
Veskovic et al. [70]	24 (9 F, 15 M); 23 ± 3.5 y; karate athletes	INT (*n* = 12): Psychological skills training CON (*n* = 12): No intervention	INT: 8 w/8 sessions; group-based and individual; 20–27 min; autogenic training/guided imagery group sessions + home training program. CON: No intervention.	CSAI-2R (score). State performance anxiety. Pre-/post-intervention.	CSAI-2R: Cognitive Anxiety: INT pre: 18.3 (2.4) CON pre: 13.1 (2.2) INT post: 16.1 (2.7) CON post: 16.7 (4.6) Somatic Anxiety: INT pre: 20.0 (4.1) CON pre: 16.3 (3.3) INT post: 18.6 (3.9) CON post: 16.2 (3.2) Self-confidence: INT pre: 18.4 (3.2) CON pre: 22.2 (3.1) INT post: 19.7 (3.9) CON post: 18.1 (2.9)	++
Kerr and Leith [63]	24 (8 F, 16 M) 18.47 ± 2.76 y; elite gymnasts	INT (*n* = 12): Psychological skills CON (*n* = 12): No intervention	INT: 32 w/16 sessions; individual; 60 min; stress management/psychological skills. CON: No intervention.	SCAT (score). Trait Performance anxiety. Pre-/post-intervention.	SCAT: INT pre: 22.93 (3.5) CON pre: 19.80 (2.8) INT post: 23.08 (4.3) CON post: 19.00 (3.1)	+
Dehghani et al. [66]	29 (29 F) 22.9 ± 0.69 y; basketball	INT (*n* = 14): Mindfulness CON (*n* = 15): No intervention	INT: 8 w/8 sessions; group-based; 90 min; mindfulness acceptance. Commitment protocol. CON: No intervention.	SCAT (score) Trait Performance anxiety. Pre-/post-intervention.	SCAT: INT pre: 18.43 (4.8) CON pre: 19.55 (4.6) INT post: 12.50 (3.8) CON post: 17.56 (3.8)	+
Fortes et al. [61]	35 (35 M) 15.93 ± 0.98 y; elite swimmers	INT (*n* = 17): Imagery CON (*n* = 18): No intervention	INT: 8 w/24 sessions; group-based; 10 min; imagery training involving videos. CON: 8 w/24 sessions; group-based; 10 min; advertisement videos.	CSAI-2R (score). State performance anxiety. Pre-/post-intervention.	CSAI-2R: Cognitive Anxiety: INT pre: 12.45 (2.1) CON pre: 12.83 (1.9) INT post: 9.07 (1.8) CON post: 12.95 (1.9) Somatic Anxiety: INT pre: 14.56 (2.1) CON pre: 14.33 (2.0) INT post: 11.18 (1.9) CON post: 14.12 (2.0) Self-confidence: INT pre: 13.49 (1.8) CON pre: 13.63 (1.9) INT post: 17.31 (1.8) CON post: 13.50 (1.9)	++
Mehrsafar et al. [73]	26 (26 M) 25.4 ± 2.4 y; wushu athletes	INT (*n* = 13): Mindfulness CON (*n* = 13): No intervention	INT: 8 w/8 sessions; group-based and individual; 10 min; mindfulness + home training program. CON: No intervention.	CSAI-2R (score). State performance anxiety. Pre-/post-intervention.	CSAI-2R: Cognitive Anxiety: INT pre: 13.09 (0.2) CON pre: 12.39 (0.2) INT post: 8.30 (0.3) CON post: 11.61 (0.3) Somatic Anxiety: INT pre: 20.77 (0.8) CON pre: 20.38 (0.8) INT post: 12.31 (0.8) CON post: 19.28 (0.8) Self-confidence: INT pre: 13.11 (0.3) CON pre: 12.41 (0.4) INT post: 16.7 (0.4) CON post: 13.13 (0.5)	++
Kanniyan [62]	36 (36 M) 16.75 ± 1.6 y; soccer athletes	INT (*n* = 18): Motivational self-talk CON (*n* = 18): No intervention	INT: 8 w/24–40 sessions; individual; 10–15 min; motivational self-talk. CON: No intervention.	CSAI-2 (score). State performance anxiety. Pre-/post-intervention.	CSAI-2: Cognitive Anxiety: INT pre: 22.6 (2.4) CON pre: 22.3 (2.1) INT post: 18.1 (2.1) CON post: 21.8 (1.9) Somatic Anxiety: INT pre: 23.8 (2.8) CON pre: 23.4 (2.9) INT post: 17.6 (1.9) CON post: 22.6 (2.1) Self-confidence: INT pre: 20.4 (3.1) CON pre: 20.5 (1.9) INT post: 24.8 (2.3) CON post: 21.2 (1.9)	+
Grobbelaar [67]	14 (14 M) 20.37 ± 1.08 y; golf athletes	INT (*n* = 7): Multimodal CON (*n* = 7): No intervention	INT: 5 w/5 sessions; group-based; 60 min; breathing, imagery, relaxation, self-talk. CON: No intervention.	CSAI-2 (score). State performance anxiety. Pre-/post-intervention.	CSAI-2: Cognitive Anxiety: INT pre: 17.63 (4.9) CON pre: 20.25 (4.1) INT post: 16.43 (4.6) CON post: 19.71 (1.4) Somatic Anxiety: INT pre: 15.38 (3.2) CON pre: 16.63 (3.5) INT post: 14.29 (4.8) CON post: 17.86 (4.6)	+
Spahn et al. [68]	21 (14 F, 7 M) 22.10 ± 2.3 y; orchestral musicians	INT (*n* = 13): Multimodal CON (*n* = 8): No intervention	INT: 14 w/14 sessions; group-based; 90 min; imagery, body awareness, breathing, psychological skills. CON: No intervention.	STAI (score). State performance anxiety. Pre-/post-intervention.	STAI State: INT pre: 50.54 (11.55) CON pre: 44.00 (10.85) INT post: 42.62 (8.24) CON post: 44.25 (14.40)	−

INT: intervention group; CON: control group; MPA: music performance anxiety; PMR: progressive muscle relaxation; CSAI-2: Competitive State Anxiety Inventory; CSAI-2R: Competitive State Anxiety Inventory Revised; HRV: heart rate variability; MPAI-A: Music Performance Anxiety Inventory: Adolescents; PAI: Performance Anxiety Inventory; SCAT: Sport Competition Anxiety Test; STAI: State-Trait Anxiety Inventory; w: week; y: years; Level of evidence: low quality (−), acceptable quality (+), and high quality (++), according to the Scottish Intercollegiate Guidelines Network Methodology checklist.

### 3.4. Intervention Characteristics

Of all included studies, the interventions of eleven studies were group-based [50,57,58,59,60,61,64,66,67,68,72], four individual [62,63,65,69], and five reported a mixed intervention form [20,49,70,71,73]. When selecting the therapy method, seven studies illustrated a multimodal intervention [20,49,57,67,68,71,72] and, thus, combined different approaches. Furthermore, three studies used mindfulness [64,66,73], motivational self-talk [50,58,62], and imagery [59,61,65]. Two studies applied psychological skills training [63,70] and one study used biofeedback [69]. One study [60] examined the effects of relaxation, mental rehearsal, and the combination of both in three intervention groups. All studies differed significantly in the number of sessions, their duration, and the total period of intervention, ranging from 1 to 24 sessions, 2 to 90 min, and 1 day to 32 weeks. One to five sessions with a total of 10–300 min of training time were conducted in seven studies [50,59,60,64,67,69,72]. Four studies used seven to eight sessions and 160–360 min [20,66,70,73], five used 14 to 40 sessions and 36–1620 min [49,61,62,63,68]. Another three studies applied a number of three to eight sessions, but gave no indication of session duration [57,58,71].

### 3.5. Outcome Measures

All studies measured at least one outcome measure for performance anxiety, whereby fifteen only considered state performance anxiety [57,58,59,60,61,62,64,65,67,68,69,70,71,72,73], four only considered trait performance anxiety [20,50,63,66], and one study considered both outcomes [49]. The CSAI-2R was used to measure state performance anxiety in seven studies [49,58,61,64,65,70,73], the CSAI-2 in four [59,60,62,67], the STAI in four, and the MPAI-A in one [57]. In two articles, the PAI was used as an additional measurement tool [69,72]. Trait performance anxiety was assessed in three articles by the SCAT [50,63,66], with one study each using the STAI [49] and MPAI-A [20]. Table 2 summarizes the effects of psychological interventions on state and trait performance anxiety in performing artists and athletes.

### 3.6. Effects of Psychological Interventions on Performance Anxiety in Athletes or Performing Artists

Figure 2 illustrates the impact of psychological interventions on measures of state performance anxiety in performing artists and athletes. Overall, the weighted mean *SMD* yielded 0.88 (*chi*^2^ = 269.17, *df* = 42, *p* < 0.001, *I*^2^ = 84%), indicating a significant large-sized effect favouring the INT groups. In addition, the sub-analyses revealed nonsignificant medium-sized effects (*SMD* = 0.62, *chi*^2^ = 4.35, *df* = 4, *p* = 0.36, *I*^2^ = 8%) for studies that used scales with total scores (i.e., MPAI-A, STAI) but significant large-sized effects (cognitive anxiety: *SMD* = 0.93, *chi*^2^ = 90.29, *df* = 12, *p* < 0.001, *I*^2^ = 87%; somatic anxiety: *SMD* = 0.92, *chi*^2^ = 99.28, *df* = 12, *p* < 0.001, *I*^2^ = 88%; self-confidence: *SMD* = 0.97, *chi*^2^ = 74.87, *df* = 11, *p* < 0.001, *I*^2^ = 85%) for studies that applied scales with sub-scores (i.e., CSAI-2). The effects of psychological interventions on measures of trait performance anxiety in performing artists and athletes are displayed in Figure 3. The weighted mean *SMD* amounted to 0.32 (*chi*^2^ = 19.07, *df* = 4, *p* = 0.0008, *I*^2^ = 79%) with the 95% confidence interval crossing zero, which indicates a nonsignificant small-sized effect in favour of the INT groups. Funnel plots are illustrated in Figure 4A,B. For psychological interventions on measures of trait performance anxiety, a symmetrical plot is shown and for those on measures of state performance anxiety, the symmetry is limited, but this is only caused by a single study (i.e., CSAI-2R (cognitive anxiety, somatic anxiety, and self-confidence) in the study of Mehrsafer et al. [73].

## 4. Discussion

To the best of our knowledge, the present systematic review with meta-analysis is the first to examine and quantify the effects of existing interventions for performance anxiety alleviation conducted in the performing arts and sports by analysing the efficacy of psychological interventions intended to enhance athletes’ anxiety coping skills. Though performance anxiety is highly prevalent in the performing arts, empirical knowledge related to this field is deficient. The present meta-analysis including 20 studies revealed mixed results with (i) large-sized effects on state anxiety and (ii) small-sized effects on trait anxiety in favour of the INT groups.

### 4.1. Effects of Psychological Interventions on Measures of State Performance Anxiety

The findings of the included studies examining state performance anxiety predominantly illustrate positive effects. The multimodal intervention approach showed mixed results, one part revealing large effects [57,68,71], the other none to low effects [49,67,72]. Duration, length, and frequency of the intervention did not seem to affect the examined outcome measures. Interventions that exclusively used either motivational self-talk [58,62] or imagery [59,61,65] as an approach consistently revealed the best results. The preferred outcomes could be related to the cognitive strategies of these interventions already known to the participants. Gregg et al. [74] illustrated that visual and motivational imagery ability increases with continuous sports practice. Especially under high psychological stress, known strategies are preferentially used and lead to superior results in emotion regulation [75] and executive function [76]. Overall, psychological interventions can reduce state performance anxiety, particularly in athletes. Further research should focus on a specific sample of athletes or performing artists and compare different kinds of intervention. This would allow targeted interventions to be developed for specific populations.

### 4.2. Effects of Psychological Interventions on Measures of Trait Performance Anxiety

The results of the included studies that addressed trait performance anxiety showed a nonsignificant small-sized effect in psychological interventions. Studies that examined performance-oriented but not elite athletes [50,66] illustrated the strongest improvements. In comparison, Clark and Willamon [49] and Osborne et al. [20] found little improvement in their study with music students, and Kerr and Leith [63] found negative effects among elite athletes. The latter could be explained by both a lack of prior questioning of the participating elite athletes concerning their experience with stress management and a general tendency for competitive athletes to benefit less from stress reduction measures than recreational athletes [77]. It should be considered that the CON group in the study by Osborne et al. [20] also received an intervention in the form of solo performance exposure, which could explain the smaller differences to the INT group. The data from Clark and Willamon [49] must also be interpreted with caution, as the CON group started the study period with significantly higher anxiety scores than the INT group and thus had greater potential for improvement. Similarly, the results of Kerr and Leith [63], who reported negative effects for the INT group compared to the CON group, may be explained by the fact that the CON group started with significantly lower scores, and therefore had lower potential for improvement. The included studies illustrate mixed results for successful interventions in terms of duration and frequency of sessions. Regarding the effectiveness of the intervention approaches, no clear preference for the treatment of trait performance anxiety can be derived from the studies. In sum, the high heterogeneity within the included studies makes it difficult to determine the efficacy of the applied interventions. In order to make evidence-based recommendations, research with higher comparability is needed. Similar to the interventions on state performance anxiety, different intervention approaches within the same population should be compared in order to more precisely determine their efficacy.

### 4.3. Limitations of the Systematic Review and Meta-Analysis

The present review’s findings should be interpreted in the context of several limitations. The heterogeneity of the included studies with regard to the implemented interventions (i.e., self-talk, imagery, mindfulness, relaxation, biofeedback, psychological skills, or multimodal), as well as number of sessions (1 to 24), session duration (2 to 90 min), total period of intervention (1 day to 32 weeks), and type of performance (level of dependence on fine motor skills) must be seen as limitations. This could be attributed to the variety of performing arts and sports disciplines and the heterogeneity of the populations. The large age difference of the participants (10.3–57.8 years) may have a significant impact on the results, as youth and adolescent athletes are generally more likely to experience severe performance anxiety [78], and psychological interventions are shown to be less effective in younger than in older participants [79].

Although interventions are limited to fine motor skills, variations in effectiveness cannot be ruled out. For instance, such variations could occur within teams of athletes (e.g., basketball) versus individual athletes (e.g., tennis), or among different performing arts and sports disciplines with varying levels of dependence on fine motor skills. Furthermore, there is considerable disagreement in the literature about the involvement of fine motor skills in different activities [6,80,81]. For example, this study excluded actors and comedians but included swimmers and rugby players because fine motor coordination was defined as part of a specific technical element. In order to determine the influence of moderator variables (e.g., intervention type) and dose–response relationships (e.g., training period), it is recommended that a direct comparison of differently designed interventions (e.g., single-mode vs. combined vs. multimodal training) or training modalities (e.g., 6 vs. 12 vs. 18 weeks of training) within a study should be conducted in the future. These significant differences lead to a limited comparability of the respective interventions. Furthermore, some studies illustrate heterogeneity in baseline anxiety levels between the CON and INT groups, creating a risk of bias that threatens the validity of the treatment effects. Although studies with substantial baseline imbalances should be excluded from meta-analyses [82], this was not added here as an exclusion criterion to comprehensively represent the state of research. However, this resulted in some included studies with low study quality ratings due to weaknesses in randomization and participant selection, which should be considered when considering the results. In order not to compromise randomization, future studies should conduct a heterogeneous treatment effect analysis after data collection to minimize the risk of bias [83]. Another limitation is that this meta-analysis intended to include studies on all groups of performing artists, but after the selection procedure, considering the inclusion and exclusion criteria as well as the methodological study quality, only studies with musicians remained among this group. According to recent reviews by Burin and Osorio [84], and Fernholtz et al. [36] on treatments for performance anxiety in musicians, cognitive behavioural therapy (CBT) is identified as the most effective psychological intervention. However, with the exception of Osborne et al. [20], all other examined studies lacked a psychological intervention in the control group, which means that a distinction between the effectiveness of individual approaches is not possible. Moreover, it should be considered that the criterion of only including open-access publications excluded many relevant research studies and therefore this meta-analysis cannot represent the full range of the literature. The aforementioned results and limitations are consistent with those of this meta-analysis but did not consider the individual components of CBT that were highlighted in our results. In addition, these studies do not distinguish between trait and state performance anxiety, which can be considered as a strength of this review.

## 5. Conclusions

The objective of the present systematic review with meta-analysis was to synthesize existing scientific evidence concerning psychological interventions from sports and the performing arts regarding their effects on performance anxiety to obtain insights beneficial for the treatment of performance anxiety. Evidence gathered via the analyses conducted in the present review proposes that psychological interventions, especially CBT, can potentially contribute to the alleviation of state performance anxiety. Based on the findings of the present meta-analysis, psychological interventions focusing on cognitive reappraisal (i.e., imagery, mindfulness) and self-talk regulation showed the most promising results. However, we conclude that further research is needed to assess the effectiveness of individual psychological interventions on specific populations in order to make targeted recommendations for practitioners. Follow-up-testing post-intervention, which was implemented in some of the reviewed studies, therefore also seems appropriate and may be recommended for prospective studies to determine whether longer-term effects of an intervention have been obtained. Furthermore, cognitive load theory proposes an overload of working memory capacity during performance anxiety, and current evidence states the essential role of working memory for visual skills that are crucial for athletic performance [85,86]. Therefore, another worthwhile aspect for future research and interventions may be examining the link between visual skills and performance anxiety. If such a link does indeed exist, training of visual skills may improve performance anxiety. Finally, the heterogeneity between studies was considerable (i.e., *I*^2^ = 79–88%). This could be corrected by further research of high methodological quality.

## Figures and Tables

**Figure 1 behavsci-13-00910-f001:**
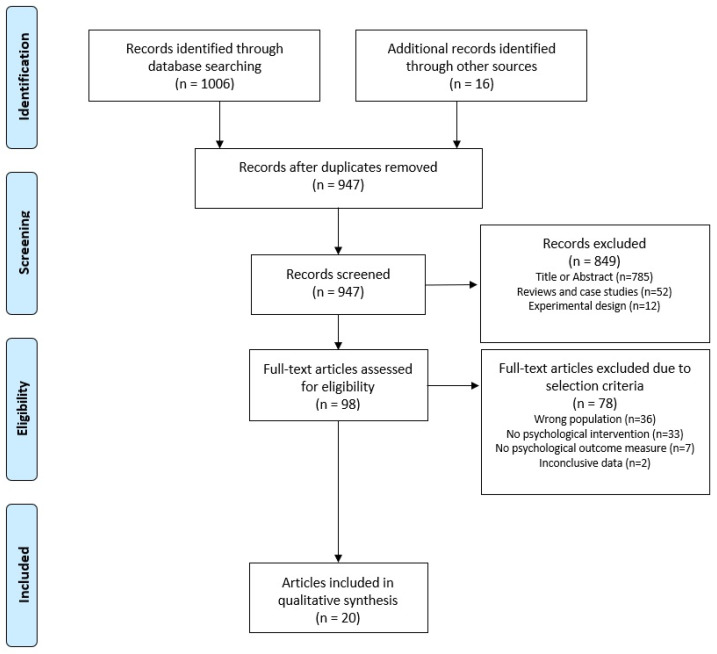
Flowchart illustrating the different phases of the literature search and study selection.

**Figure 2 behavsci-13-00910-f002:**
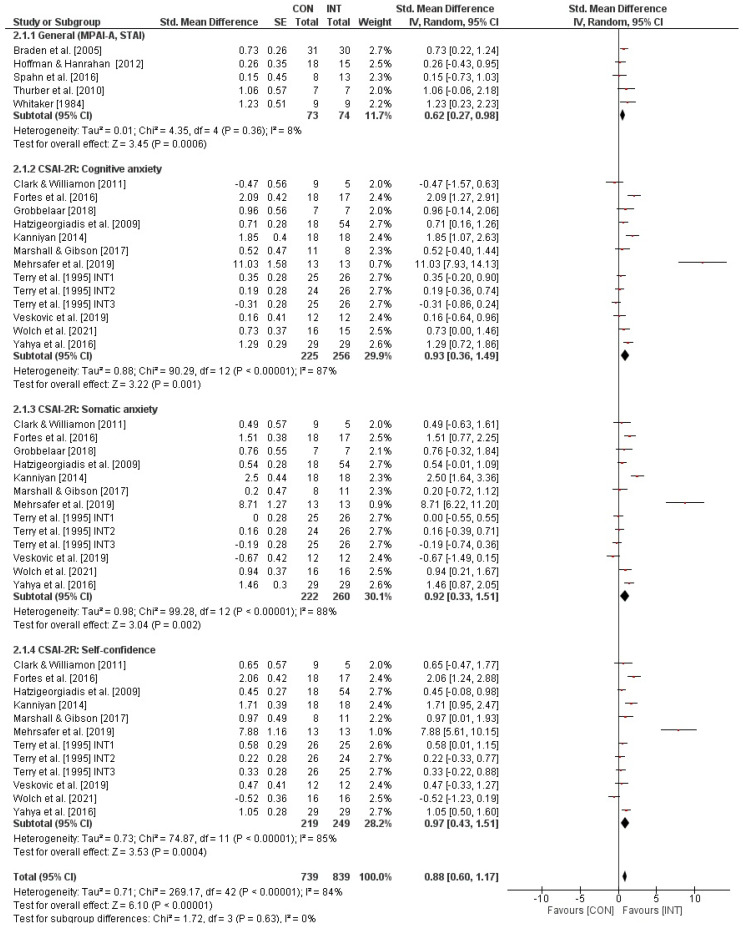
Effects of psychological interventions on measures of state performance anxiety (e.g., CSAI-2) in performing artists and athletes [49,57,58,59,60,61,62,64,65,67,68,69,70,71,72,73]. *CI*: confidence interval; *CON*: control group; *df*: degrees of freedom; *INT*: intervention group; *IV*: inverse variance; *SE*: standard error; *Std.*: standard.

**Figure 3 behavsci-13-00910-f003:**
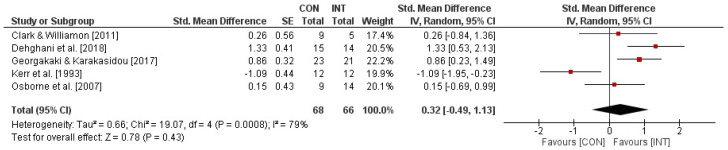
Effects of psychological interventions on measures of trait performance anxiety (e.g., SCAT) in performing artists and athletes [20,49,50,63,66]. *CI*: confidence interval; *CON*: control group; *df*: degrees of freedom; *INT*: intervention group; *IV*: inverse variance; *SE*: standard error; *Std.*: standard.

**Figure 4 behavsci-13-00910-f004:**
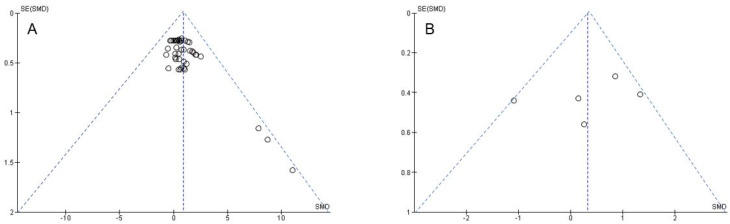
Funnel plots for publication bias assessment regarding (**A**) state and (**B**) trait performance anxiety. *SE*: standard error; *SMD*: standardized mean difference.

**Table 1 behavsci-13-00910-t001:** Overview of the preferred and alternative outcome by category.

Category	Preferred Outcome	Alternative Outcome
State performance anxiety	Competitive State Anxiety Inventory 2 Revised score (CSAI-2R; *n* = 7)	State-Trait Anxiety Inventory state score (STAI; *n* = 4). Competitive State Anxiety Inventory 2 score (CSAI-2; *n* = 4). Music Performance Anxiety Inventory for Adolescents score—State anxiety (MPAI-A; *n* = 1). Performance Anxiety Inventory score (PAI; *n* = 2).
Trait performance anxiety	Sport Competition Anxiety Test score (SCAT; *n* = 3)	State-Trait Anxiety Inventory trait score (*n* = 1). Music Performance Anxiety Inventory for Adolescents score—Trait anxiety (MPAI-A; *n* = 1).

## Data Availability

The original contributions presented in the study are included in the article/Appendix A; further inquiries can be directed to the corresponding author.

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
