# Peer review of "Effects of Psychological Interventions on Performance Anxiety in Performing Artists and Athletes: A Systematic Review with Meta-Analysis"

_behavsci, 2023, doi:10.3390/bs13110910_

Round 1
Reviewer 1 Report
Comments and Suggestions for Authors
First of all I would like to congratulate the authors for their article entitled Effects of psychological interventions on performance anxiety in performing artists and athletes: a systematic review with meta-analysis which deals with different psychological interventions and their effect on anxiety in athletes and artists. The study is very interesting and the subject is increasingly necessary to study.
However, there are a couple of issues that I would like to see clarified by the authors.
1. Some self-plagiarism has been detected in the abstract as well as in the materials and methods section (Effects of Physical Training on Physical and Psychological Parameters in Individuals with Patella Tendinopathy: A Systematic Review and Meta-Analysis; Effects of athletic training on physical fitness and stroke velocity in healthy youth and adult tennis players: A systematic review and meta-analysis). We strongly recommend to modify the affected paragraphs.
I would also like to know if the authors consider that the large age difference between the studies participants could affect the results. In my opinion, this is a major limitation.
Finally, although it is recorded in the limitations of the study, the great diversity of variables (types of athletes, types of psychological techniques used ...) it provides a lack of clarity that could be corrected and not only be justified by its inclusion in the justification of the manuscript.
Author Response
Author's Reply to the Review Report (Reviewer 1):
Comments and Suggestions for Authors
Comment: First of all I would like to congratulate the authors for their article entitled Effects of psychological interventions on performance anxiety in performing artists and athletes: a systematic review with meta-analysis which deals with different psychological interventions and their effect on anxiety in athletes and artists. The study is very interesting and the subject is increasingly necessary to study.
However, there are a couple of issues that I would like to see clarified by the authors.
Some self-plagiarism has been detected in the abstract as well as in the materials and methods section (Effects of Physical Training on Physical and Psychological Parameters in Individuals with Patella Tendinopathy: A Systematic Review and Meta-Analysis; Effects of athletic training on physical fitness and stroke velocity in healthy youth and adult tennis players: A systematic review and meta-analysis). We strongly recommend to modify the affected paragraphs.
Response: We would like to thank the reviewer for the detailed comments on our work, which we have tried to implement appropriately in the subsequent sections. Changes were highlighted in yellow.
We apologize for this and changed the respective sentence appropriately.
- 1, line 19-21: Abstract: „A systematic search of the literature according to the PRISMA guidelines was conducted on the databases PubMed, Medline, SPORTDiscus, PsycInfo, Embase, and Web of Science from 01 January 1960 to 09 November 2022.”
Comment: I would also like to know if the authors consider that the large age difference between the studies participants could affect the results. In my opinion, this is a major limitation.
Response: Thank you for this valuable comment. We agree and have included this point in the Limitation section and also provided justification for the importance of age.
- 19, line 366-369: Discussion section: „The large age difference of the participants (10.3–57.8 years) may have a significant impact on the results, as youth and adolescent athletes are generally more likely to experience severe performance anxiety [1], as well as psychological interventions are shown to be less effective in younger than in older participants [2].”
Comment: Finally, although it is recorded in the limitations of the study, the great diversity of variables (types of athletes, types of psychological techniques used ...) it provides a lack of clarity that could be corrected and not only be justified by its inclusion in the justification of the manuscript.
Response: We agree with the review and acknowledged this aspect as follows:
- 20, line 426-428: Conclusions section: “Finally, the heterogeneity between studies was considerable (i.e., I2 = 79–88%). This could be corrected by further research of high methodological quality.”
References
- Rocha, V.; Osório, F. Associations between Competitive Anxiety, Athlete Characteristics and Sport Context: Evidence from a Systematic Review and Meta-Analysis. Archives of Clinical Psychiatry (São Paulo) 2018, 45, 67–74, doi:10.1590/0101-60830000000160.
- Hut, M.; Glass, C.R.; Degnan, K.A.; Minkler, T.O. The Effects of Mindfulness Training on Mindfulness, Anxiety, Emotion Dysregulation, and Performance Satisfaction among Female Student-Athletes: The Moderating Role of Age. Asian Journal of Sport and Exercise Psychology 2021, 1, 75–82, doi:10.1016/j.ajsep.2021.06.002.
- Nordin-Bates, S.M.; Quested, E.; Walker, I.J.; Redding, E. Climate Change in the Dance Studio: Findings from the UK Centres for Advanced Training. Sport, Exercise, and Performance Psychology 2012, 1, 3–16, doi:10.1037/a0025316.
- Sutani, S.; Akutsu, T. The Life History of Performance Anxiety in Japanese Professional Orchestral Players: A Case Series. Medical Problems of Performing Artists 2019, 34, 63–71, doi:10.21091/mppa.2019.2009.
- Thomson, P.; Jaque, S.V. Attachment and Childhood Adversity in Athletes, Actors, Dancers, and Healthy Controls. International Journal of Sport and Exercise Psychology 2019, 17, 334–349, doi:10.1080/1612197X.2017.1349824.
- Mellalieu; Hanton, S.; Fletcher, D. A Competitive Anxiety Review: Recent Directions in Sport Psychology Research. In A Competitive Anxiety Review: Recent Directions in Sport Psychology Research; 2006; pp. 1–77.
- Baumeister, R.F. Choking under Pressure: Self-Consciousness and Paradoxical Effects of Incentives on Skillful Performance. Journal of Personality and Social Psychology 1984, 46, 610–620, doi:10.1037/0022-3514.46.3.610.
- Mellalieu; Hanton, S.; O’Brien, M. Intensity and Direction of Competitive Anxiety as a Function of Sport Type and Experience. Scand J Med Sci Sports 2004, 14, 326–334, doi:10.1111/j.1600-0838.2004.00389.x.
- Magill, R.A.; Anderson, D. Motor Learning and Control: Concepts and Applications; Tenth edition.; McGraw-Hill: New York, NY, 2014; ISBN 978-0-07-802267-8.
- Schmidt, R.A.; Lee, T.D.; Winstein, C.J.; Wulf, G.; Zelaznik, H.N. Motor Control and Learning: A Behavioral Emphasis; Sixth edition.; Human Kinetics: Champaign, IL, 2019; ISBN 978-1-4925-4775-4.
Reviewer 2 Report
Comments and Suggestions for Authors
Thank you for the opportunity to review this interesting study. I will begin with a short summary of how I interpret the paper, followed by specific comments.
This study is a systematic review aiming to explore whether psychological interventions can influence state and trait performance anxiety outcomes. Results seem to suggest that state anxiety can be influenced by some psychological intervention, but trait anxiety is unclear. The authors discuss the lack of homogeneity in their studies, which is appreciated to make it clear that the literature is lacking in this respect.
Firstly, I would like to congratulate the authors on a well-designed and thought-out study. I believe that there are definite questions that come about as a result of this study, and in that sense the aim is fulfilled. I do have questions regarding the inclusion criteria (see comment 4 below) in that a reliance on fine motor skills may not be a straightforward criterion and I feel it may need more explanation. My main critique in the results is the lack of discussion of the confidence intervals in the std mean difference in the trait anxiety results. It seems this is an important aspect to mention so the reader does not interpret this as being a significant result.
Introduction:
1. Line 92 (“As for athletes…”) – I’m not sure I understand this sentence. There is quite a lot of discussion of performing artists, and the authors seem to be implying that athletic performance is in line with this (with which I agree); however, I am confused by the wording of this, as is sounds as though the athletes are separate from the intensive training demands referenced.
2. Line 134 - Wording “assess[es]”.
3. Lines 137-139 - Is there any operationalization of what “relevant” insights and “promising” questions might be?
Methods:
4. Lines 157-158 – How was the dependence of fine motor skill performance defined? In terms of sport, there could be those that argue for the need of fine motor skills in running in order to maintain balance, for example, despite the predominance of gross motor skills as the authors claim on lines 102-104. One included study in this systematic review is with rugby players, which could be argued to predominately use gross motor skills. On lines 224-225 the authors mention that 36 studies examined populations not relevant to the aims of this review, so I wonder what exactly the criteria are for excluding these articles (if this is what is meant by relevance). I think having a clearer definition would be beneficial to the reader to fully understand what is meant by this.
Results:
5. Lines 285-287: While the effect size may be small, the 95% CI still crosses 0, so this is still a non-significant std. mean difference?
Discussion
6. Lines 323-325: I may have missed it in the results somewhere, but I cannot find results that support this, as I see no comparisons between performance type (i.e. artists or athletes). Is this an analysis which was performed?
7. Lines 331-332: Is this still a conclusion that can be drawn without mentioning the lack of statistical significance (i.e. 95% CI includes the 0)? It seems important to mention this; the authors do discuss heterogeneity in the limitations section, but it still reads as though the small effect size is significant.
Author Response
Author's Reply to the Review Report (Reviewer 2):
Comments and Suggestions for Authors
Comment: Thank you for the opportunity to review this interesting study. I will begin with a short summary of how I interpret the paper, followed by specific comments. This study is a systematic review aiming to explore whether psychological interventions can influence state and trait performance anxiety outcomes. Results seem to suggest that state anxiety can be influenced by some psychological intervention, but trait anxiety is unclear. The authors discuss the lack of homogeneity in their studies, which is appreciated to make it clear that the literature is lacking in this respect. Firstly, I would like to congratulate the authors on a well-designed and thought-out study. I believe that there are definite questions that come about as a result of this study, and in that sense the aim is fulfilled. I do have questions regarding the inclusion criteria (see comment 4 below) in that a reliance on fine motor skills may not be a straightforward criterion and I feel it may need more explanation. My main critique in the results is the lack of discussion of the confidence intervals in the std mean difference in the trait anxiety results. It seems this is an important aspect to mention so the reader does not interpret this as being a significant result.
Response: We would like to thank the reviewer for the detailed comments on our work, which we have tried to implement appropriately in the subsequent sections. Changes were highlighted in turquoise.
Comment: Introduction: 1. Line 92 (“As for athletes…”) – I’m not sure I understand this sentence. There is quite a lot of discussion of performing artists, and the authors seem to be implying that athletic performance is in line with this (with which I agree); however, I am confused by the wording of this, as is sounds as though the athletes are separate from the intensive training demands referenced.
Response: We thank the reviewer for this comment and have made the beginning of the sentence a little more explicit to clarify that we equate athletes in sports, as well as performing artists in this context.
- 2, line 92-94: Introduction section: “Similar to athletes in sports, this broad array of demands requires musicians, actors, and dancers to devote themselves to intensive training to meet and exceed these standards, generally from early childhood onwards [3–5]”
Comment: 2. Line 134 - Wording “assess[es]”.
Response: Thank you for this correction, which we have implemented accordingly.
- 3, line 135-139: Introduction section: “This current research gap serves as the rationale for this paper. This systematic review and meta-analysis assesses the current state of the art on the effects of psychological interventions used in sports and the performing arts to reduce performance anxiety and to determine their effects on performance anxiety in performing artists and athletes.”
Comment: 3. Lines 137-139 - Is there any operationalization of what “relevant” insights and “promising” questions might be?
Response: Thank you for this comment. We have elaborated on it and revised the text accordingly.
- 3, line 139-142: Introduction section: „It aims to provide relevant insights for the treatment and prevention of performance anxiety in the performing arts. In addition, promising research questions from related scientific evidence will be identified to encourage and facilitate interdisciplinary approaches.”
Comment: Methods: 4. Lines 157-158 – How was the dependence of fine motor skill performance defined? In terms of sport, there could be those that argue for the need of fine motor skills in running in order to maintain balance, for example, despite the predominance of gross motor skills as the authors claim on lines 102-104. One included study in this systematic review is with rugby players, which could be argued to predominately use gross motor skills. On lines 224-225 the authors mention that 36 studies examined populations not relevant to the aims of this review, so I wonder what exactly the criteria are for excluding these articles (if this is what is meant by relevance). I think having a clearer definition would be beneficial to the reader to fully understand what is meant by this.
Response: Thank you for this valuable comment on improving the clarity of our inclusion criteria. The aim was to exclude professional groups such as actors, as they are not comparable to the physical performance demands of athletes in sports. We have made this particular statement more specific in various sections. However, we agree with the reviewer about the heterogeneity of the definition in the literature and consider this as an important limitation. Therefore, we have added this in the limitations section as well.
- 3, line 104-108: Introduction section: „In contrast, this sympathetic overactivation might be perceived as debilitating by golfers and musicians as well as any athletes or performing artists relying on fine motor skills due to its potential to impede motor skill accuracy. Specifically, it could impair motor accuracy during these movements, as fine motor skills require much more cognitive resources [6,7].”
- 4, line 159-170: Materials and Methods section: “We used the following criteria to determine eligibility: (a) Population: athletes or performing artists depending on fine motor skill performance including a technical component; (b) Intervention: psychological interventions; (c) Comparator: active or passive control group (i.e., different psychological intervention, no training at all); (d) Outcome: at least one measure of anxiety; (e) Study design: controlled trials with pre- and post-measures. The exclusion criteria were as follows: (a) participants’ discipline was not dependent on fine motor skill performance or without a technical component (e.g., actors); (b) reported data did not allow for calculation (i.e., no central tendency and dispersion measure provided in the results section or upon request); (c) effects were examined without control condition; (d) assessments did not include a psychological outcome measure; (e) cross-sectional study design.”
- 15, line 228-231: Results section: “Thirty-six studies examined a study population not relying on fine motor skills (e.g., actors/comedians), or a mixed population including those. Another 33 studies did not use a psychological intervention, seven did not report any psychological outcome measures, and two did not provide conclusive data.”
- 19, line 373-377: Discussion section: Furthermore, there is considerable disagreement in the literature about the involvement of fine motor skills in different activities [8–10]. For example, this study excluded actors and comedians but included swimmers and rugby players because fine motor coordination was defined as part of a specific technical element.”
Comment: Results: 5. Lines 285-287: While the effect size may be small, the 95% CI still crosses 0, so this is still a non-significant std. mean difference?
Response: We thank the reviewer for this comment and have better clarified the results.
- 17, line 289-291: Results section: “The weighted mean SMD amounted to 0.32 (Chi2 = 19.07, df = 4, p = 0.0008, I2 = 79%) with the 95% confidence interval crossing zero, which indicates a non-significant small-sized effect in favour of the INT-groups.”
Comment: Discussion: 6. Lines 323-325: I may have missed it in the results somewhere, but I cannot find results that support this, as I see no comparisons between performance type (i.e. artists or athletes). Is this an analysis which was performed?
Response: We apologize for this misleading statement, which we have deleted in the revised version.
Comment: 7. Lines 331-332: Is this still a conclusion that can be drawn without mentioning the lack of statistical significance (i.e., 95% CI includes the 0)? It seems important to mention this; the authors do discuss heterogeneity in the limitations section, but it still reads as though the small effect size is significant.
Response: We fully agree with the reviewer and have mentioned the lack of significance in the Discussion section and adjusted the Conclusion for the trait anxiety outcomes.
- 18, line 333-334: Discussion section: “The results of the included studies which addressed trait performance anxiety showed a non-significant small-sized effect in psychological interventions.”
- 20, line 411-413: Conclusion section: “Evidence gathered via the analyses conducted in the present review proposes that psycho-logical interventions, especially CBT, can potentially contribute to the alleviation of state performance anxiety.”
References
- Rocha, V.; Osório, F. Associations between Competitive Anxiety, Athlete Characteristics and Sport Context: Evidence from a Systematic Review and Meta-Analysis. Archives of Clinical Psychiatry (São Paulo) 2018, 45, 67–74, doi:10.1590/0101-60830000000160.
- Hut, M.; Glass, C.R.; Degnan, K.A.; Minkler, T.O. The Effects of Mindfulness Training on Mindfulness, Anxiety, Emotion Dysregulation, and Performance Satisfaction among Female Student-Athletes: The Moderating Role of Age. Asian Journal of Sport and Exercise Psychology 2021, 1, 75–82, doi:10.1016/j.ajsep.2021.06.002.
- Nordin-Bates, S.M.; Quested, E.; Walker, I.J.; Redding, E. Climate Change in the Dance Studio: Findings from the UK Centres for Advanced Training. Sport, Exercise, and Performance Psychology 2012, 1, 3–16, doi:10.1037/a0025316.
- Sutani, S.; Akutsu, T. The Life History of Performance Anxiety in Japanese Professional Orchestral Players: A Case Series. Medical Problems of Performing Artists 2019, 34, 63–71, doi:10.21091/mppa.2019.2009.
- Thomson, P.; Jaque, S.V. Attachment and Childhood Adversity in Athletes, Actors, Dancers, and Healthy Controls. International Journal of Sport and Exercise Psychology 2019, 17, 334–349, doi:10.1080/1612197X.2017.1349824.
- Mellalieu; Hanton, S.; Fletcher, D. A Competitive Anxiety Review: Recent Directions in Sport Psychology Research. In A Competitive Anxiety Review: Recent Directions in Sport Psychology Research; 2006; pp. 1–77.
- Baumeister, R.F. Choking under Pressure: Self-Consciousness and Paradoxical Effects of Incentives on Skillful Performance. Journal of Personality and Social Psychology 1984, 46, 610–620, doi:10.1037/0022-3514.46.3.610.
- Mellalieu; Hanton, S.; O’Brien, M. Intensity and Direction of Competitive Anxiety as a Function of Sport Type and Experience. Scand J Med Sci Sports 2004, 14, 326–334, doi:10.1111/j.1600-0838.2004.00389.x.
- Magill, R.A.; Anderson, D. Motor Learning and Control: Concepts and Applications; Tenth edition.; McGraw-Hill: New York, NY, 2014; ISBN 978-0-07-802267-8.
- Schmidt, R.A.; Lee, T.D.; Winstein, C.J.; Wulf, G.; Zelaznik, H.N. Motor Control and Learning: A Behavioral Emphasis; Sixth edition.; Human Kinetics: Champaign, IL, 2019; ISBN 978-1-4925-4775-4.
Round 2
Reviewer 1 Report
Comments and Suggestions for Authors
The authors have made the suggested changes, accepting the manuscript as follows